# Wi-CAL: A Cross-Scene Human Motion Recognition Method Based on Domain Adaptation in a Wi-Fi Environment

Zhanjun Hao [1,2,*][ID], Juan Niu [1], Xiaochao Dang [1,2] and Danyang Feng [1]

1   College of Computer Science and Engineering, Northwest Normal University, Lanzhou 730070, China
2   Gansu Province Internet of Things Engineering Research Center, Lanzhou 730070, China
*   Correspondence: haozhj@nwnu.edu.cn

**Abstract:** In recent years, research on Wi-Fi sensing technology has developed rapidly. This technology automatically senses human activities through commercial Wi-Fi devices, such as lying down, falling, walking, waving, sitting down, and standing up. Because the movement of human parts affects the transmission of Wi-Fi signals, resulting in changes in CSI. In the context of indoor monitoring of human health through daily behavior, we propose Wi-CAL. More precisely, CSI fingerprints were collected at six events in two indoor locations, and data enhancement technology Dynamic Time Warping Barycentric Averaging (DBA) was used to expand the data. Then the feature weighting algorithm and convolution layer are combined to select the most representative CSI data features of human action. Finally, a classification model suitable for multiple scenes was obtained by blending the softmax classifier and CORrelation ALignment (CORAL) loss. Experiments are carried out on public data sets and the data sets before and after the expansion collected in this paper. Through comparative experiments, it can be seen that our method can achieve good recognition performance.

**Keywords:** device-free sensing; human motion recognition; CSI; domain adaptation; data enhancement





## 1. Introduction

With the rapid update and iteration of information technology, the Internet of things has ushered in a new Artificial Intelligence and Internet of things (AIOT) era. As a critical technology of human-computer interaction in intelligent life, situational awareness has entered people's daily life [1–4]. Through the collection and analysis of scene information, this technology enables computing devices to have an intelligent perception of the environment and personnel information in the perception space. According to the collection method of human action samples, action recognition mainly includes the following three categories: (1) vision-based [5]; (2) sensor-based [6]; (3) based on Wi-Fi [7,8]. In a vision-based system, human motion is captured by using the optical camera; in the sensor-based system, the limb motion characteristics are captured by sensors on the human body (such as sports bracelets). Through the collected human action information, different data processing processes, and classification learning, the above method can recognize the action information of personnel. However, these technologies have their limitations. The vision-based perception scheme has some defects, such as profound influence by light, poor privacy, blind spots, etc. At the same time, long-term detection also has high requirements for energy consumption. It is difficult to wear and deploy sensors for the sensor-based sensing scheme and is not generally simple.

Due to the universality of Wi-Fi devices and their all-weather, non-contact characteristics, Wi-Fi has become one of the main tools for indoor motion perception [9,10]. At the same time, CSI is also gradually known by everyone. Because it can abandon the presentation of RSS single data and provide more abundant channel information, in recent years, CSI is mainly used to study indoor user positioning [11], user identification [12], and user behavior identification [13,14]. This paper studied situational perception based

on CSI signals in Wi-Fi, focusing on human motion recognition and providing higher perception accuracy.

Indoor personnel health monitoring is crucial in people's growing living needs. People in indoor environments, especially the elderly, children, and the disabled, need to monitor their daily behaviors, which is conducive to health data monitoring. Wi-Fi activity awareness technology can reduce security risks such as privacy disclosure compared to other real-time health monitoring technologies [15], so we perceived some daily behaviors of indoor personnel to explore the effectiveness of the identification method proposed in our paper.

There are many common characteristics between CSI and time-series data. Therefore, this paper attempted to learn from some expansion methods of time series data. Because the most prominent data expansion method of time series data is Dynamic Time Warping (DTW), we used a data enhancement method, DTW Barycentric Averaging (DBA), based on DTW distance. DBA can be used to enhance small-scale data sets. Weighted DTW distance was applied to various time series data sets. Reference [16] proved that DBA could effectively expand time series data. Therefore, we used this method to expand the CSI action data collected in the scene to alleviate the overfitting of the model caused by the too-small data set.

Reference [17], in the context of monitoring the daily activities of the elderly, used the transceiver characteristics of wireless signals to aggregate the CSI streams of the same receiving antenna and used the method of deep learning for training, verified on the public data set and the data set collected by ourselves, and the average recognition rate could reach more than 95%. Reference [18] designed a long short-term memory convolutional neural network framework that recognizes different activities through CSI from commercial Wi-Fi devices. Reference [19] combines the amplitude and phase of CSI to design two classifiers and realizes a unique action recognition method through the DTW algorithm and support vector machine (SVM) model, with an average recognition effect of 98%. However, the generalization and migration capabilities of models in these methods are relatively weak, so this paper aimed to improve the model migration capabilities and reduce the differences in model recognition rates in different scenarios.

Because the same model produces significant differences in the effect of two similar tasks, there is a domain shift between the data distribution of different tasks. Similarly, in the action recognition task based on CSI, the action recognition model built by the data set collected in one scene does not have a good recognition rate in other scenes. Therefore, this paper established a knowledge transfer structure from the source domain to the target domain by combining the relevant knowledge of domain adaptation and crossing the distribution differences between the source domain and target domain [20]. The purpose was to build a model that could learn domain invariant features, through which we can explicitly reduce the domain shift, to enhance the feature transfer ability of action recognition tasks. The method of correlation alignment loss can solve the problem of domain adaptation in the field of image recognition. It is to align the distribution of source domain features and target domain features through second-order statistics to reduce the gap between the two domains. Therefore, we used this method to achieve the domain adaptation of the model in our manuscript.

The contributions of our study are:

(1) Due to the small amount of data in the actual data collection process, it is easy to cause the model to overfit, so the data enhancement method based on DTW distance was used to expand the CSI data, including human behavior, to reduce the error of the classification model;

(2) By combining feature selection algorithms, different action recognition tasks pay more attention to the features strongly related to actions to reduce the interference of static objects on the collected CSI data;

(3) A cross-scene human motion recognition model based on domain adaptation is proposed, which can reduce the differences between CSI data in different scenes. Additionally, the recognition accuracy can also reach more than 90%. Experiments on existing data sets verify the effectiveness of this method.

In Section 2, we described the related technology and Wi-CAL. In Section 3, we introduced experiments and discussed the results. In Section 4, we summarized our work.

## 2. Method

Due to the complex environment of Wi-Fi coverage in real life, the premise that the standard CSI sample data collected under experimental conditions are consistent with the actual CSI sample distribution is damaged, which leads to the reduced accuracy of the single recognition model for human movement recognition in different environments. In order to improve the recognition model, the deviation problem was put forward based on the domain of invariant features depth migration study human action recognition method, based on the analysis of the characteristics related to aligning loss under different environment conditions combined with neural network to realize the distribution differences between human action recognition and typical scenario, the domain of the adaptive model. The rough structure of the model is shown in Figure 1.

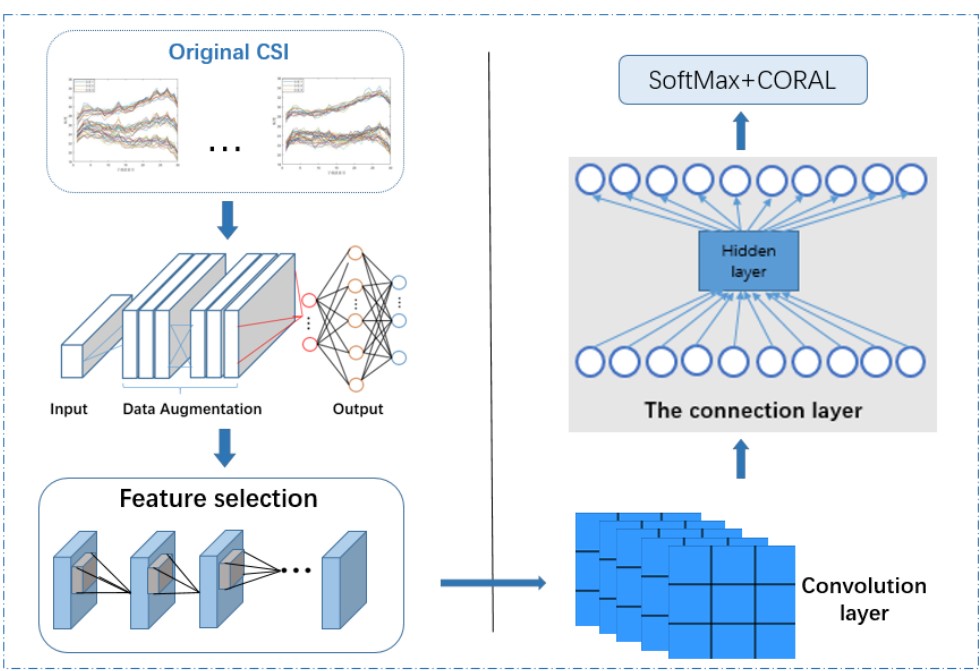

**Figure 1.** Overview of Wi-CAL.

### 2.1. Data Calibration

Due to the large CSI noise from the commercial wireless network card, the main reason is divided into two parts: internal and external factors. Internal factors are the internal state changes in Wi-Fi network card hardware, such as transmission rate adaptation, transmission power change, CSI reference level change, etc. External factors are static interference caused by environmental differences and dynamic effects caused by changes in indoor activities of different people. These internal state transitions cause high amplitude pulses and burst noise in CSI flow. Data calibration is to reduce noise as much as possible. Hampel filter, band-pass filter, and principal component analysis (PCA) were used to suppress useless signal components [21].

Hampel Filtering: When collecting CSI motion information, due to the influence of hardware or environmental noise, the collected CSI information often has abnormal information, also known as outliers. The initially collected information is shown in Figure 2a. Outliers interfere with the extraction of human motion features, so they must be eliminated. Hampel filtering can find outliers in CSI data sequence by decision-making and replace their outliers with characteristic values that are more representative of the series to denoise [22]. CSI data sets arriving at the Hampel filter were given a hypothetical distribution and probability model, and the data were processed by using an inconsistency test according to the hypothesis. When the data are greater than the discrimination threshold set by Hampel, the data points are considered outliers and eliminated.

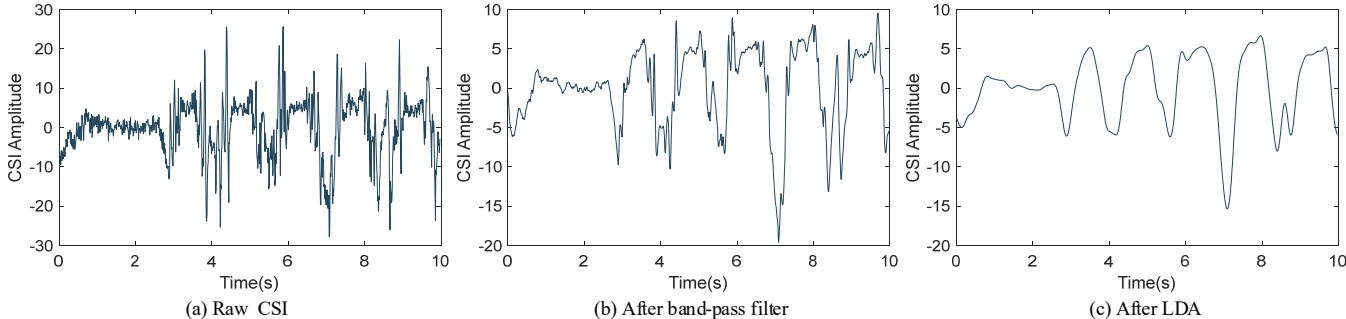

**Figure 2.** Data calibration.

Butterworth Band-pass Filtering: Because the design method of the Butterworth filter has a slowly decreasing gain in the stopband, which makes its attenuation speed slower than other types of filters, it does not distort the characteristic information generated by human activities. In order to retain the characteristics of human motion and eliminate the interference of high-frequency noise, adjust the cut-off frequency and order of the filter. After noise reduction by a band-pass filter, CSI retains human motion information to the greatest extent.

Linear Discriminant Analysis (LDA): As a compelling data preprocessing method, LDA can extract features well. By projecting high-dimensional samples into the best vector space, it can realize data feature extraction and feature space dimension compression. It can ensure that after projection, the differences within the same kind of samples are reduced, and the differences between different types of pieces are expanded. Unlike PCA, LDA can make better use of prior knowledge to preliminarily classify and reduce the dimension of CSI information of human actions, providing more accurate and effective motion CSI information for subsequent motion feature extraction.

### 2.2. Data Enhancement

As the amount of data greatly influences the training of the model, in general, the larger the amount of data, the better the training effect of the model. Therefore, this study also used the DBA algorithm to expand the data set collected in the two environments.

The background technology of DBA is as follows:

DBA is a weighted form of DTW center averaging technique that creates an infinite number of new time series from a given set of time series by varying the weights. Reference [23] adopted a weighting method called the average selection method and proved its effectiveness. The algorithm is actually to find the consistent sequence (average sequence) of the sequential data set $S$. Let us say the tuple of one is $S = \{s1, s2, \ldots sn\}$, when the elements in $S$ are real numbers, it is easy to find the average of $S$:

$$\text{avg} = \frac{s1 + s2 + \ldots + sn}{n} \tag{1}$$

We extended and shortened the two CSI sequences to obtain the shortest distance measure between the two CSI sequences. Therefore we need to find a way to choose a path to minimize the final distance. Therefore, we can use the principle that a point in a CSI sequence corresponds to multiple points in another CSI sequence and then use the matrix method to calculate the shortest path according to the cost of each element of the sequence to align two similar time-series better and generate a new average line, to expand the CSI data set. The above content is the core idea of DTW.

Normally, two or more sequences are computed to generate a new sequence, but the computation is always performed in pairs. Therefore, if we have two sequences, $A = \{a_1, a_2, \ldots, a_m\}, B = \{b_1, b_2, \ldots, b_n\}, m > n$. Then, the Euclidean distance is used to calculate the distance between every two points of each sequence, $D(a_i, b_j)$, where $1 \leq i \leq m, 1 \leq j \leq n$, as shown in the matrix Figure 3 below.

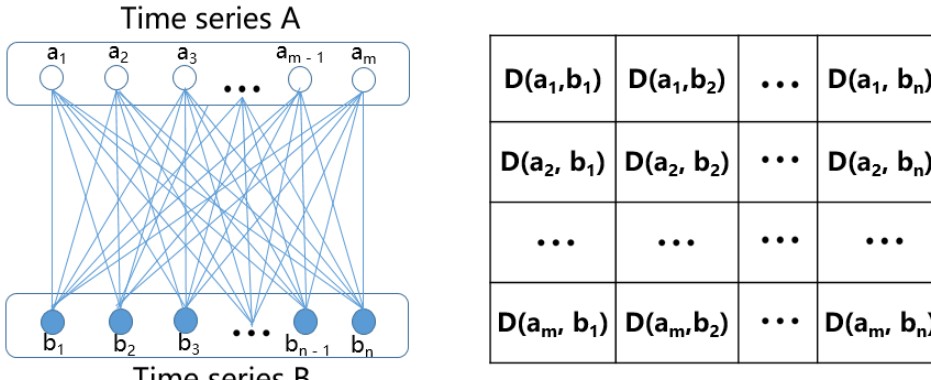

**Figure 3.** Calculation method and calculation matrix.

Therefore, in our study, the general steps of the DBA algorithm are summarized as follows:

1. We calculated the DTW between each sequence and the temporary average sequence to be refined to find the correlation between the average series' coordinates and the sequence set coordinates;
2. In the first step, we updated each coordinate of the average sequence to the center of gravity of its associated coordinates.

We supposed that $R = \{R_1, \ldots, R_N\}$ is a set of sequences to be averaged and $D = \langle D_1, \ldots, D_T \rangle$ is the average sequence of iterations for the *i*th time. $D' = \langle D'_1, \ldots, D'_T \rangle$ is the update of $D$ at iteration $i + 1$. In addition, each coordinate of the mean sequence is defined in Euclidean space $E$:

$$\forall t \in [1, T], D_t \in E \tag{2}$$

We use function *assoc* to connect each coordinate of the average sequence to one or more coordinates of the R sequence. This function is computed when performing a DTW calculation between each sequence of $D$ and R. Then, the *t*th coordinate of the average sequence $D'_t$ is defined as Equation (3) [23]:

$$D'_t = barycenter(assoc(D_t)) \tag{3}$$

where $barycenter\{X_1, \ldots, X_\alpha\} = \frac{X_1 + \ldots + X_\alpha}{\alpha}$. Where the addition of $X_i$ is vector addition, Algorithm 1 describes the complete DBA calculation in detail.

---

**Algorithm 1 DBA**

---

Require: $D = \langle D_1, \ldots, D_{T'} \rangle$ Initial average sequence
Require: $R_1 = \langle r_1, \ldots, r_{1_T} \rangle$ The first sequence to average

$\vdots$

Require: $R_n = \langle r_{n_1}, \ldots, r_{n_T} \rangle$ The $n$th sequence to be averaged
Let $T$ be the length of sequences
Let *assocT* ab be a table of size $T'$ containing in each cell a set
  of coordinates associated with each coordinate of $D$
Let m[$T$, $T$] be a temporary DTW (cost,path) matrix
   *assocT* ab $\leftarrow [0, \ldots, 0]$
  for *seq* in R do
    $m \leftarrow DTW(D, seq)$
    $i \leftarrow T'$
    $j \leftarrow T$
    while $i \geq 1$ and $j \geq 1$ do
      *assocT* ab[$i$] $\leftarrow$ *assocT* ab[$i$] $\cup$ *seq j*
      $(i, j) \leftarrow second(m[i, j])$
    end while
  end for
  for $i = 1$ to $T$ do
$D'_i$ = barycenter(*assocT* ab[$i$]) {
see Equation (3)
}
end for
return $D'$

---

### 2.3. Feature Extraction and Feature Selection

Before the data analysis, we needed to extract useful attribute information from the data and use it as a data feature to analyze such data. For a specific learning algorithm, it is unknown which feature is effective. Therefore, it is necessary to select relevant features beneficial to the learning algorithm from all features.

Figure 4 shows the general feature selection process. First, the original CSI feature set needs to generate a candidate subset. We used an evaluation function to compare the generated subset with the previously generated subset, leaving the better ones, and then used a stop standard to judge whether the feature selection process is stopped. Finally, we verified the effectiveness of the selected feature subset.

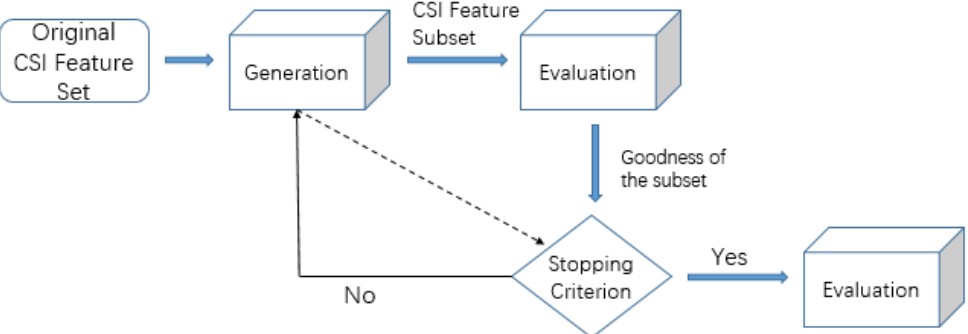

**Figure 4.** Rough flow chart of feature selection.

In the method of human motion recognition based on Wi-Fi, we also needed to extract and select the features of CSI data related to motion. CSI sample data include time-domain characteristics and frequency-domain characteristics. The time-domain parts are mean, maximum, minimum, skewness, kurtosis, variance, and average line passing rate, which can capture the pattern of the CSI waveform. Frequency domain features are normalized entropy, normalized energy, and FFT peaks of measured energy distribution patterns. These

ten similar features play an essential role in Wi-Fi-based activity recognition. Not all features are favorable for recognition. Therefore, our paper used ReliefF to calculate the weights of different feature categories and the voting mechanism to accurately select valuable features to identify human movements. ReliefF assigns weights to each characteristic attribute based on its recognition effect [24]. Algorithm 2 describes the complete ReliefF calculation in detail.

---

**Algorithm 2 ReliefF**

---

Require: Weight of each feature *T*.
Let all *T* be 0;
for *i* = 1 to m do
Randomly select a sample *R*;
Find the nearest neighbor sample H of the same category as *R*;
    Find the nearest neighbor sample m of different categories of *R*;
    for *A* = 1 to *N* do
W(*A*) = W(*A*) − diff(*A,R,H*)/*m* + diff(*A,R,M*)/*m*;
for *A* = 1 to *N* do
If W(*A*) ≥ δ
Add the *A*th feature to *T*
end

---

where $m$ is the number of samples sampled, and the threshold of feature weight is $\delta$. Finally, all features are weighted. Such as Equations (4) and (5) [24].

$$W(A) = W(A) - diff(A, R, H)/m \\ + diff(A, R, M)/m \tag{4}$$

$$\sum_{C \in \text{class}(R)} \frac{\left[ \frac{p(C)}{1-p(Class(R))} \sum_{j=1}^{k} diff\left(A, R, M_j(C)\right) \right]}{(mk)} \tag{5}$$

In which $A = 1 \ldots N$, $m$ is the number of samples. The weights of FFT value, normalized energy, and normalized entropy calculated by the above algorithm account for 43%, 35%, and 22%, respectively. It can be learned that different feature categories have different weights. Specifically, FFT peaks have a relatively high weight.

The feature projection neural network layer uses the weights obtained by ReliefF as prior knowledge and automatically focuses the effective features as a continuous training recognition model. Specifically, multiple feature projection layers transform the subspace of three category features into unified feature space in parallel. The feature projection layer consists of a $1 \times 1$ convolutional network and a gated linear unit. The projected feature subspace is multiplied by the weights of each corresponding feature category, and these weights are calculated by averaging the weights of all feature attributes. Therefore, the three feature data sets ($l = 3$) are converged into a unified feature space as shown in Equation (6) [14]:

$$H'_{\text{input}} = [x_1, x_2, \ldots, x_l] \tag{6}$$

Where $[x_1, x_2, \ldots, x_l]$ represents the weighted sum of feature maps generated in feature projection layer $1 \ldots l$. $H'_{\text{input}}$ is the unified feature space.

### 2.4. Activity Recognition

#### 2.4.1. Domain Adaptation Method Based on Divergence

When two similar tasks had different data distributions, we needed to apply the training model of one study to another. At that time, we needed to use a unique migration learning mechanism called domain adaptation (DA). There are three basic techniques of

field adaptation: the discrepancy-based DA approach, confrontation-based, and reconstruction based.

In classification problems, convolutional neural networks (CNN) can make very accurate predictions if high-quality, annotated training data are used for training. However, if a particular problem does not have a particularly large annotated data set, in this case, we can use the relevant knowledge of transfer learning to train the data set of a similar problem network and then tune the network with the small labeled data set we collected. Moreover, if there is a significant difference between test data and training data during model training, the model performance may not be very good. In this case, domain adaptation can greatly help by training different (but related) target data through models trained on the source data and achieving high classification accuracy. Therefore, the previous association between the source and target domains determines the quality of domain adaptation.

Based on Discrepancies in both fields, the DA method assumes that fine-tuning the deep network model with labeled or unlabeled target data can reduce the difference between the two domains. We used domain adaptation based on divergence to solve the problem of domain difference in the human motion recognition model based on CSI. The domain adaptation method using the correlation alignment principle is shown in Figure 5.

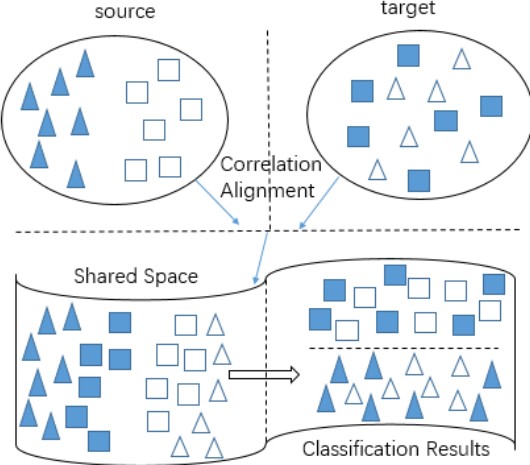

**Figure 5.** Domain adaptation principle.

Our study used the divergence-based domain adaptation method called the CORAL loss to solve the problem of the distribution difference between the source and target domains. CORAL [20] aligns the source and target characteristics distribution in an unsupervised manner. It relies on linear transformation to minimize the square Frobenius norm distance of the covariance of source and target characteristics. CORAL must first calculate the second-order statistics (covariance) between the source and target data and then convert the source domain into the target domain to align their distribution. For the converted source domain data, we need to train additional classifiers, such as SVM, and finally classify the target domain data set.

### 2.4.2. Action Recognition Method Based on Domain Adaptation

In this study, the category classifier mainly classifies the input data through the full connection layer and Softmax function. In the full connection layer, the paper used the correlation alignment algorithm to measure the distribution difference between two data sets to maximize the model's migration ability. We added the L2 norm to the full connection layer of the convolutional neural network so that our model can more effectively alleviate the overfitting phenomenon and improve the accuracy while increasing the number of full connection layers. The domain classifier mainly comprises a full connection layer and cross-state classifier. In this part, gradient inversion was used to maximize the classification loss of the domain classifier; that is, in the process of error backpropagation, gradient

descent can be prevented by a reverse gradient to maximize the classification loss of the domain classifier. The specific framework of the model is shown in Figure 6.

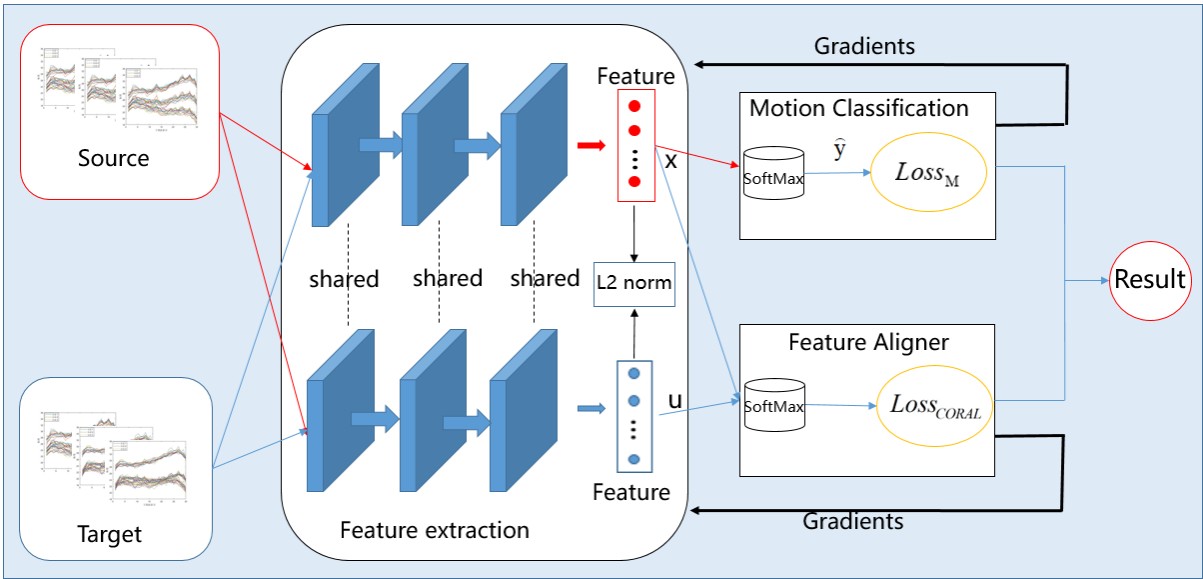

**Figure 6.** Model-specific framework.

As shown in Figure 6, we divided the same action CSI data collected in different scenarios into source data and target data at first; that is, the domain in our study refers to the scene. Then, the features of the source data set and the target data set were extracted through the feature extractor and shared. At this time, the motion classification was used to classify the actions to obtain the prediction result $\widehat{y}$, and then the loss $Loss_{\mathrm{M}}$ between the prediction result and the real result $y$ needed to be calculated.

$$Loss_{\mathrm{M}} = -\frac{1}{|x|} \sum_{i=1}^{|x|} \sum_{C=1}^{C} y_{iC} \log\left(\widehat{y}_{iC}\right) \tag{7}$$

where $C$ is the number of action data, and $x$ is the input data.

In order to address the problem of the universality of different models in different data domains, this paper leveraged deep features and tag source data that were pre-trained on large data domains. At the same time, we wanted the features we eventually learned to work well in the target domain. The first goal could be achieved by initializing network parameters from a generic pre-training network and fine-tuning the tag source data. For the second goal, we proposed to minimize the difference of second-order statistics between source and target feature activation, that is, to minimize $Loss_{CORAL}$. Therefore, the feature aligner guarantees domain independence through this principle. The working principle is as follows:

We suppose that the source dataset labeled $L_s = \{y_i\}, i \in \{1, \ldots, L\}$ is $D_s = \{x_i\}, x \in R^d$, and the unmarked target dataset is $D_T = \{u_i\}, u \in R^d$.

Assume that the amount of data in the source dataset and in the target dataset is $n_S$. $x$ and $u$ are the learnable $d$-dimensional deep activation $\phi(I)$ of input $I$. We supposed that $D_S^{ij}(D_T^{ij})$ is the $j$th dimension of the $i$th source (target) data, and $C_S$ ($C_T$) represents the characteristic covariance matrix. CORAL loss is the distance between the second-order statistic (covariance) of the source feature and the target feature, as shown in the following Equation (8):

$$Loss_{CORAL} = \frac{1}{4d^2} \|C_S - C_T\|_F^2 \tag{8}$$

where $\|.\|_F^2$ represents square matrix Frobenius norm. The covariance matrix of source data and target data is shown in Equations (9) and (10) [20]:

$$C_S = \frac{1}{n_S - 1}\left(D_S^{\mathrm{T}}D_S - \frac{1}{n_s}\left(1^{\mathrm{T}}D_S\right)^{\mathrm{T}}\left(1^{\mathrm{T}}D_S\right)\right) \tag{9}$$

$$C_T = \frac{1}{n_T - 1}\left(D_T^{\mathrm{T}}D_T - \frac{1}{n_T}\left(1^{\mathrm{T}}D_T\right)^{\mathrm{T}}\left(1^{\mathrm{T}}D_T\right)\right) \tag{10}$$

where 1 is the column vector where everything is equal to 1. Then compute the gradient of the input feature by using the chain rule [20]:

$$\frac{\partial Loss_{CORAL}}{\partial D_S^{ij}} = \frac{1}{d^2(n_S - 1)}\left(\left(D_s^{\mathrm{T}} - \frac{1}{n_S}\left(1^{\mathrm{T}}D_S\right)^{\mathrm{T}}1^{\mathrm{T}}\right)^{\mathrm{T}}(C_S - C_T)\right)^{ij} \tag{11}$$

$$\frac{\partial Loss_{CORAL}}{\partial D_T^{ij}} = \frac{1}{d^2(n_T - 1)}\left(\left(D_T^{\mathrm{T}} - \frac{1}{n_T}\left(1^{\mathrm{T}}D_T\right)^{\mathrm{T}}1^{\mathrm{T}}\right)^{\mathrm{T}}(C_S - C_T)\right)^{ij} \tag{12}$$

We input the feature vectors of the two domains into the feature aligner and then calculated $Loss_{CORAL}$ to calculate the degree of difference between the two domains. Finally, we optimized our output results through gradient inversion and the parameters of the network so as to minimize both $Loss_M$ and $Loss_{CORAL}$, and then we shared the parameters obtained with the minimum loss in the training network and the test network so as to obtain a domain adaptive network model. Finally, the result was obtained by repeatedly updating the model, that is, the recognition result of the motion.

## 3. Discussion

### 3.1. Experiment Setting

We used two laptops equipped with Intel 5300 network cards, one as a transmitter and connected to a set of antennas, and the other as a receiver connected to a set of three antennas. The two laptops were tuned into a CSI-enabled platform for CSI data collection. In the experiment, we used the Ubuntu16.04LTS operating system, and the laptops of both computers were Intel corei3-4150. The experiment included six types of indoor human behavior, including wave hand (wh), walk (w), fall (f), lie down (ld), sit down (sd), and stand up (su).

The following four experiments were carried out in two experimental environments (hall and laboratory): A. The original setting of the room, in which an experimenter performs six activities in the test area, and the environment was not changed. B. A volunteer performed six activities in the test area, but the layout and equipment location constantly changed during the data collection process. Tables and chairs as obstacles were placed at different positions in the test area, and the positions of tables and chairs were moved and rotated to create environmental dynamics. C. When one volunteer performed activities, other volunteers walked, chatted, or waved near the test area. D. Based on scene C, we added distractors in the environment.

Figure 7 shows a variety of data acquisition scenarios in the conference room environment. Figure 7a shows ① and ② data acquisition scenes at different positions in the conference room, respectively. Figure 7b shows the scene where there are static people in the room except the experimental personnel and the position of indoor equipment changes. Figure 7c shows people walking indoors; the location of indoor equipment was changed, and indoor equipment was added. Figure 7d shows the explanation of the previous three plans.

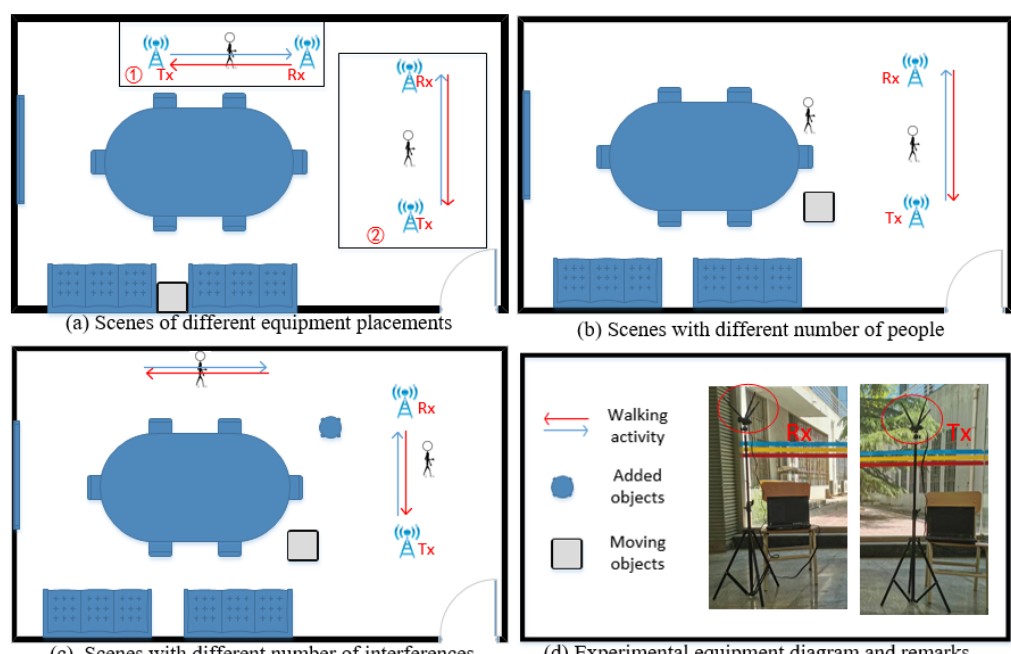

**Figure 7.** Experimental environment plan and experimental equipment.

In order to fully understand the changes in user diversity, we recruited six experimenters (three men and three women). In total, more than 3100 CSI samples were collected in two experimental sites (hall and conference room), then these data were enhanced by data expansion technology, and then the amplitude information of the CSI signal was extracted for further data processing. During acquisition, the system received 114 subcarriers of each CSI data. The input size of our CSI data is $3 \times 114 \times 500$. Table 1 shows the number of motion CSIs collected by six volunteers.

**Table 1.** Experimental data collected by six experimenters in two experimental environments.

| Sence | | Person 1 | Person 2 | Person 3 | Person 4 | Person 5 | Person 6 |
|---|---|---|---|---|---|---|---|
| Lobby | A | 80 | 80 | 80 | 80 | 80 | 80 |
| | B | 60 | 60 | 60 | 60 | 60 | 60 |
| | C | 60 | 60 | 60 | 60 | 60 | 60 |
| | D | 60 | 60 | 60 | 60 | 60 | 60 |
| Meeting Room | A | 80 | 80 | 80 | 80 | 80 | 80 |
| | B | 60 | 60 | 60 | 60 | 60 | 60 |
| | C | 60 | 60 | 60 | 60 | 60 | 60 |
| | D | 60 | 60 | 60 | 60 | 60 | 60 |

### 3.2. Experimental Factor Analysis

3.2.1. Overall Accuracy

We used ordinary Wi-Fi devices in this section to evaluate our method. We tested the performance of these systems in four modes, A, B, C, and D, in the laboratory and conference room. Tagged data from A were used as source domain data, and unlabeled data from B, C, or D test environment settings were used as target domain data. Figure 8 shows the overall perceived recognition rate of six daily behaviors in the laboratory and conference room.

It can be seen from Figure 8 that the average recognition accuracy of our model can reach more than 89.36% in that environment. These encouraging results show the superiority of Wi-CAL in different scenarios.

|           | wave hand | walk | fall | lie down | sit down | stand up |
|-----------|-----------|------|------|----------|----------|----------|
| wave hand | 0.92 | 0.02 | 0.02 | 0.02 | 0.01 | 0.01 |
| walk      | 0.02 | 0.93 | 0.03 | 0.02 | 0.00 | 0.00 |
| fall      | 0.01 | 0.03 | 0.91 | 0.02 | 0.03 | 0.00 |
| lie down  | 0.01 | 0.00 | 0.03 | 0.93 | 0.03 | 0.00 |
| sit down  | 0.01 | 0.01 | 0.02 | 0.01 | 0.94 | 0.01 |
| stand up  | 0.01 | 0.01 | 0.02 | 0.03 | 0.01 | 0.92 |

**Figure 8.** Comprehensive evaluation experiment.

### 3.2.2. Effect of CORrelation ALignment

Since we adopted the divergence-based domain adaptation method CORAL, in order to verify the effect of this method on our study, a comparative experiment was specially designed. Figure 9 shows the ROC curves of Wi-CAL and Wi-CAL without CORAL.

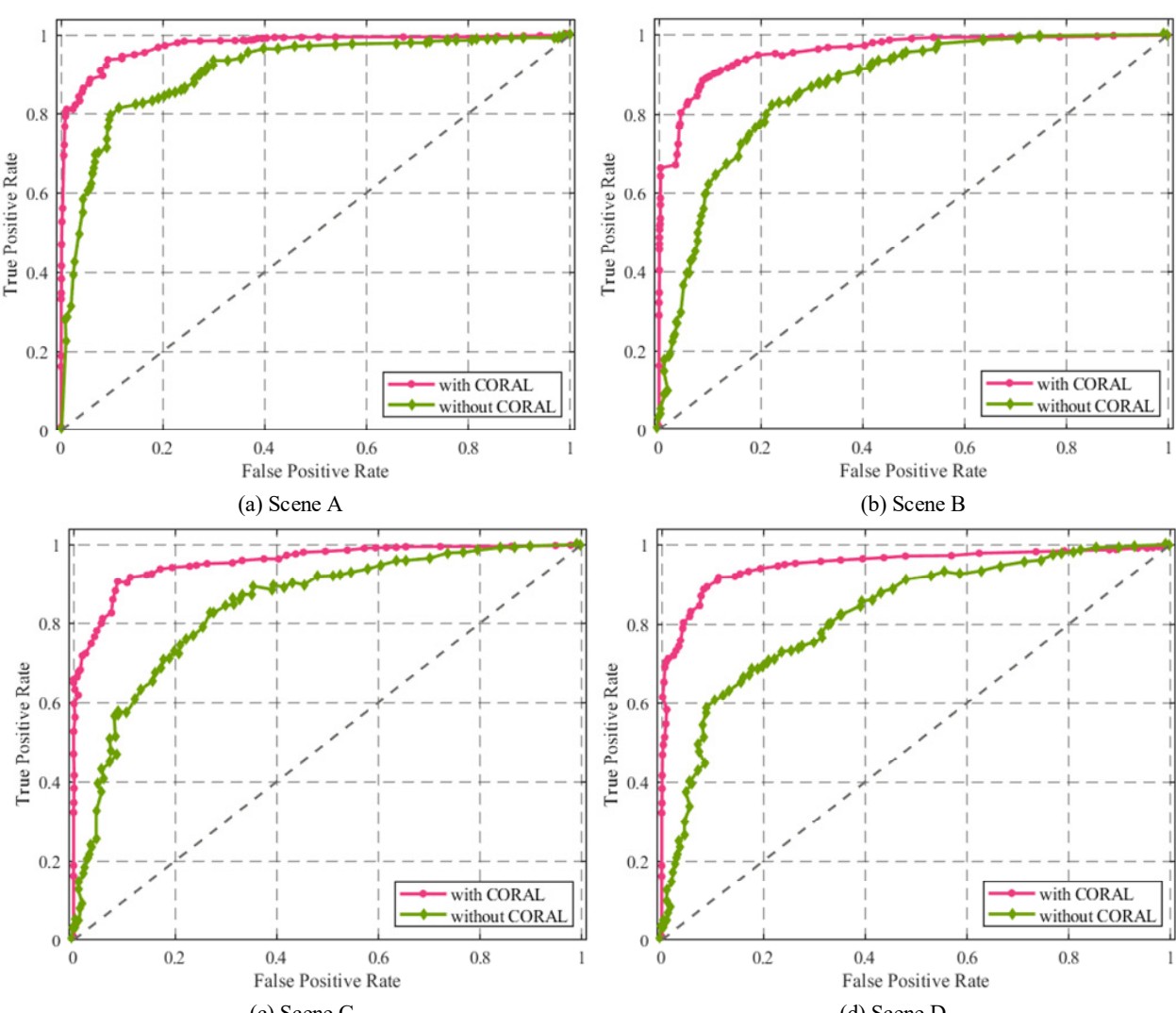

(a) Scene A

(b) Scene B

(c) Scene C

(d) Scene D

**Figure 9.** Correlation alignment recognition effect when used and not used.

In Figure 9, the area enclosed by the red curve and X axis is the AUC area when the CORAL method was used in Wi-CAL (with CORAL), and the green curve and X axis are the AUC area when the CORAL method was not used (without CORAL). We can observe that in the diagram without CORAL, the ROC curve and AUC area of A, B, C, and D scenes have a great dissimilarity, which is in line with the real situation. Because the number of distractors increases in turn in scenes B, C, and D, the multipath effect is more complex, and the signal interference is more and more serious, so the model recognition effect decreases with the increase in the number of people and distractors. However, the model recognition rate in scene A is the highest because it has the simplest layout and stable multipath effect. At the same time, we can also see that when using the coral method, the ROC curves obtained from the four different scenes have little difference, and the AUC area is also similar, indicating that the model recognition effect is similar, thus indicating the effectiveness of using the coral method in cross scene recognition in our study.

### 3.2.3. Effect of Data Augmentation

In order to verify the contribution of DBA to the model in this paper, the following verification was made in the experimental part, especially for whether data augmentation is helpful in improving the recognition rate of the human motion recognition model. We randomly selected the data sets of the two movements in the experiment for the experiment. Figure 10 shows the recognition effect of walk (w) and sit down (sd) with and without data enhancement.

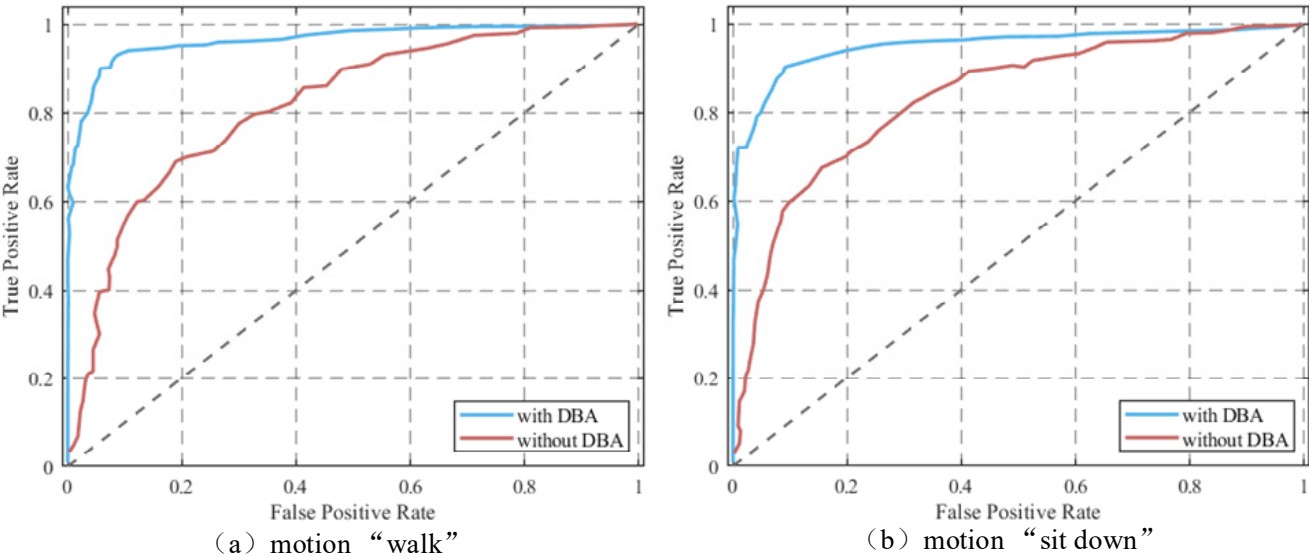

**Figure 10.** Influence of DBA technology on model recognition results when used or not used.

From the ROC curve and AUC area in Figure 10, we can see that no matter what kind of motion, when we use DBA technology, the recognition effect of the action data set model collected in this paper was significantly higher. It can be seen from Figure 10 that DBA technology has a certain contribution to improving the model recognition rate of Wi-CAL because we believe that the data enhancement technology was effective, and we will use the data enhancement technology to improve the model recognition rate in the subsequent experiments.

### 3.2.4. Effect of Distance between Transmitter and Receiver

The distance between devices affects not only the sensing range but also the sensing recognition results. Four groups of spacing correlation experiments were set up in the open hall to explore the influence of spacing on recognition resolution. In the experiment, RX and TX were set at intervals of 1 m, 2 m, 3 m, and 4 m, respectively. Under each spacing,

the experimenter kept the same position to perform the setting action. The cumulative distribution functions (CDF) of the error rate are shown in Figure 11.

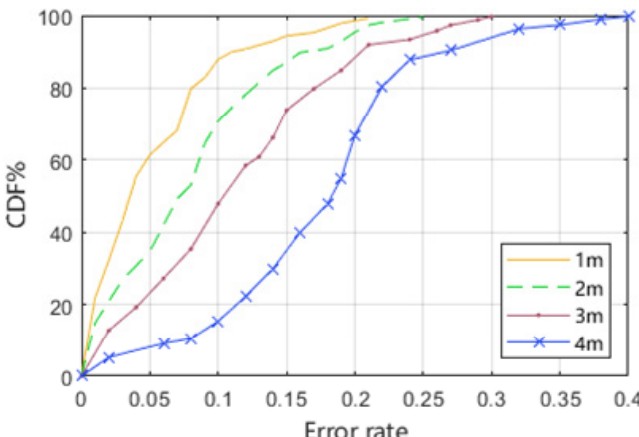

**Figure 11.** CDF of error rate.

The X-axis represents the error rate in the activity behavior recognition results, and the Y-axis represents the cumulative distribution function percentage of the error rate. When the distance between the transmitter and receiver is 1 m, the closer the curve is to the upper left, the better the recognition effect of human activity behavior is, and the error rate of nearly 80% is less than 0.1. Looking back at the CDF curve, when the equipment spacing is 4 m, the error rate of nearly 51% is less than 0.2. The influence of distance on the recognition accuracy is proved; that is, with the increase in the distance between the two devices, the recognition accuracy declines. For Wi-CAL, the deployment interval is set to 1 m, which is similar to the recognition ability of 2 m, and the deployment interval of 2 m provides greater human perception, so the experimental interval is initially set to 2 m. On the whole, Wi-CAL can still maintain a good recognition resolution in the 4 m sensing area, which meets the general needs of scene sensing space.

### 3.2.5. Effect of Sampling Rates

The sampling frequency of CSI data specifically impacts the model recognition rate, so we set different sampling frequencies to verify. We changed the sampling frequency by changing the sampling time interval. At the same time, to reduce the accidental parameters that lead to the weak generalization ability of the model due to the arbitrary division of the training set and verification set, we choose cross-validation ten times to avoid this situation.

Figure 12 shows the recognition accuracy of Wi-CAL under different sampling rates. It can be seen that when the sampling time interval increases from 5 ms to 10 ms, the average recognition accuracy decreases from 94.71% to 90.13%. When the sampling time interval increases to 20 milliseconds, the average recognition accuracy decreases to 80.50%, and when it increases to 25 ms, the average recognition accuracy decreases to less than 75%. Therefore, we learned from it that when the sampling interval is no more than 20 ms, Wi-CAL obtains a better recognition accuracy. Our experiments show that samples with a high sampling rate have richer action information; therefore, higher accuracy can be obtained. However, in practical experiments, with the improvement of the sampling rate, the recognition system also encountered the problem of a high packet loss rate. Therefore, by using data sets collected at low sampling rates, Wi-CAL still has high recognition accuracy.

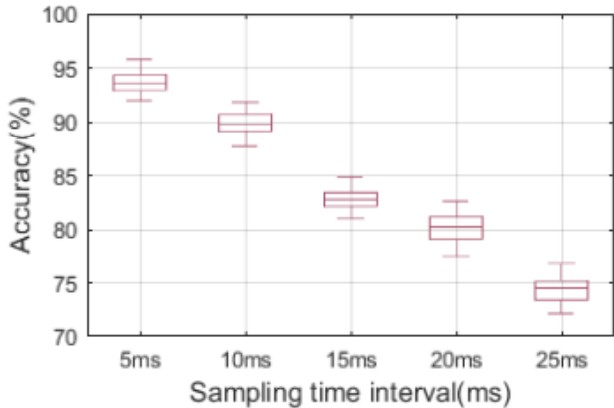

**Figure 12.** Different sampling time intervals.

### 3.2.6. Validation of Different Data Sets

In order to prove the effectiveness of our method, we conducted experiments based on the public data set ARIL [25]. ARIL provides CSI data of six gestures (i.e., Hand Up, Hand Down, Hand Left, Hand Right, Hand Circle, and Hand Cross) and two corresponding labels, one indicating the type of activity and the other indicating the position. We used the hand-related data set collected in this paper to train the model and the public data set to test.

Figure 13 shows the model identification results of the new data as the test set. The test results of Figure 13 show that the proposed model Wi-CAL can achieve automatic domain adaptation, and the test results can reach more than 88%. It shows that the model in this paper migrates to two different domains.

| | Hand-Up | Hand-Down | Hand-Left | Hand-Right | Hand-Circle | Hand-Cross |
|---|---|---|---|---|---|---|
| Hand-Up | 0.90 | 0.03 | 0.03 | 0.02 | 0.01 | 0.01 |
| Hand-Down | 0.02 | 0.91 | 0.04 | 0.03 | 0.00 | 0.00 |
| Hand-Left | 0.02 | 0.04 | 0.88 | 0.02 | 0.04 | 0.00 |
| Hand-Right | 0.02 | 0.00 | 0.03 | 0.91 | 0.04 | 0.00 |
| Hand-Circle | 0.02 | 0.02 | 0.02 | 0.02 | 0.91 | 0.01 |
| Hand-Cross | 0.02 | 0.01 | 0.02 | 0.04 | 0.01 | 0.90 |

**Figure 13.** Wi-CAL on public data sets.

### 3.2.7. Comparison of Different Classification Algorithms

In order to evaluate the performance of the classification method in the Wi-CAL model, in the experimental environment of this paper, the CSI data of six subjects sitting down (su) were collected as experimental samples, and six subjects acted in four scenarios: A, B, C, and D. the overall results we designed were compared with common motion classification algorithms such as hidden Markov model (HMM) [26], support vector machine (SVM) [27], and random forest (RF) [28] in the previous work. The action recognition effect of each classification method is shown in Figure 14a. In addition, we introduced the current deep learning algorithm gated recurrent unit (GRU) [29], long short-term memory (LSTM) [30], and recurrent neural network (RNN) [31], which is strongly adapted to time series, and compare it with Wi-CAL, as shown in Figure 14b.

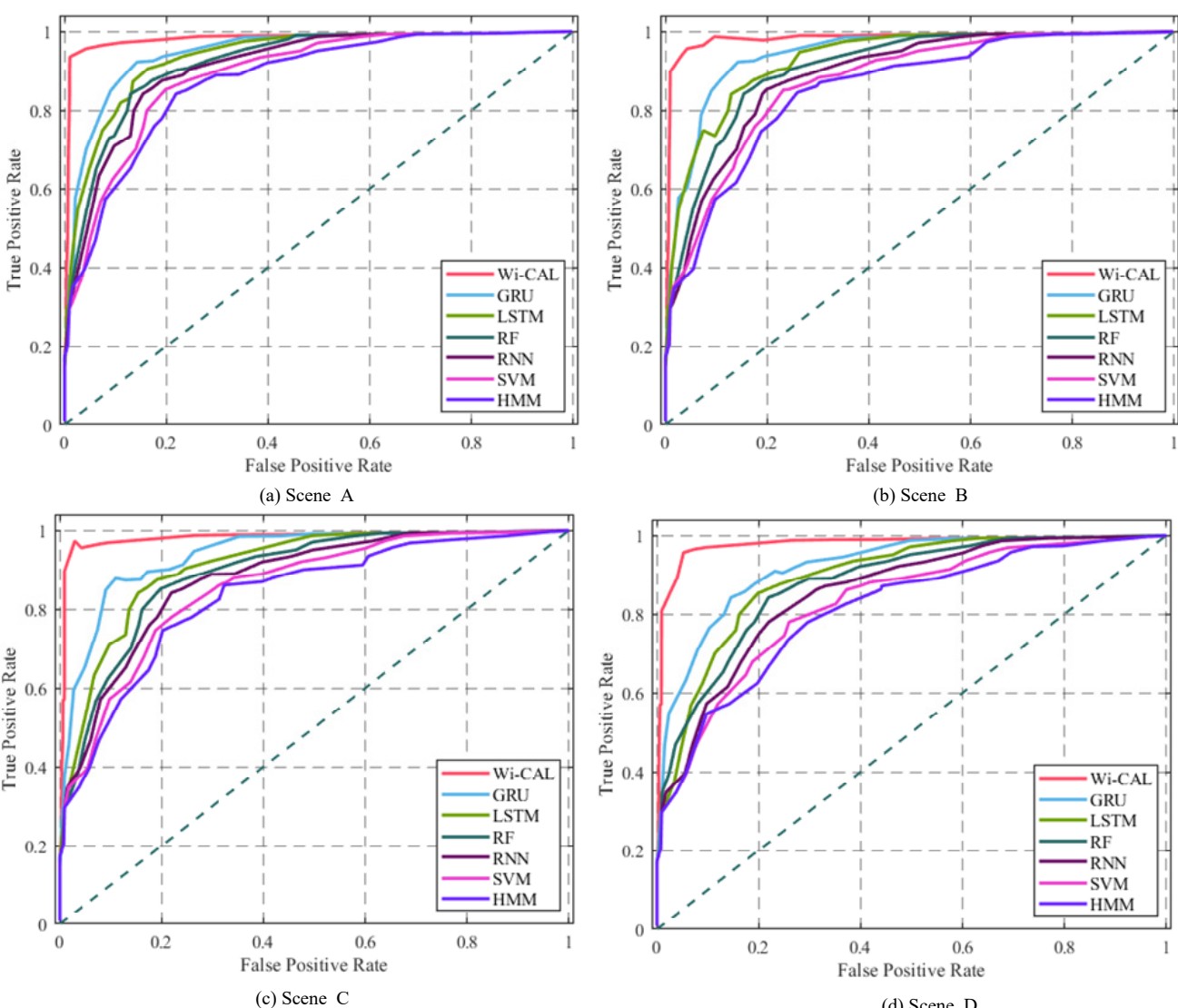

**Figure 14.** Different classifier algorithms.

From the ROC curve in Figure 14, it can be clearly found that in scene A, the accuracy rates of the Wi-CAL, HMM, SVMm, and RF are 94.56%, 73.10%, 76.2%, and 85.3%, respectively, indicating that the improved neural network deep learning method can achieve better results than the traditional method in recognition and classification after feature extraction. Compared with the basic neural network structures GRU, LSTM, and RNN, the most prominent feature of Wi-CAL is cross-scene recognition, and the recognition rate does not change much with the scene change. At the same time, compared with the other three deep learning algorithms, Wi-CAL has better adaptability and robustness to human motion characteristics. It can be seen from the four figures that the recognition effect of Wi-CAL does not change much when the four scenes are switched, but the recognition rate of the other six algorithms changes greatly, which fully proves the effectiveness of this method in solving the problem that the model performance changes with the scene switching.

### 3.2.8. Comparison of Different Motion Recognition Methods

In order to verify the superiority of Wi-CAL for passive human motion recognition, we also compared it with the model with a high motion recognition rate in recent years. Table 2 details the differences between the five comparison methods and Wi-CAL in the following five aspects.

**Table 2.** Comparison of recognition accuracy of each model.

| Project | Activity | Algorithm | Feature | Average Accuracy | Advantage | Disadvantage |
|---|---|---|---|---|---|---|
| Wi-CAL | wave hand, walk, fall, lie down, sit down, and stand up | DBA, ReliefF, and CORAL | CSI Amplitude | 93.57% | The migration capability of models in different scenarios is realized. | The migration capability of the model in more environments is not discussed. |
| WiNum [32] | Gesture number "1–9" | DTW and SVM | CSI Amplitude and phase | 91.06% | It improves the utilization of CSI and effectively recognizes handwritten digits. | There are too few application scenarios, and the actual use is limited. |
| MCBAR [33] | running, walking, falling down, boxing, circling arms, and cleaning floor | Generative Adversarial Networks | CSI Amplitude | 90.79% | It overcomes the performance degradation of different environment models. | Unlabeled data cannot be effectively used to solve the degradation of model performance. |
| ABLSTM [34] | Lie down, Fall, Walk, Run, Sit down, and Stand up | Bidirectional long short term memory neural network | CSI Amplitude and phase | 90.24% | It can focus on more representative features, making feature learning richer information. | Data labeling is difficult, and unmarked data cannot be used efficiently. |
| Sheng et al. [35] | bend, box, clap, pull, throw, and wave | Deep CNN | CSI Amplitude and phase | 88.85% | It can learn higher-level features and make full use of CSI information. | The system does not discuss the recognition performance under various scenarios and cannot judge the practicability of the system. |
| Wi-SL [36] | 12 sign language actions | K-means, Bagging, and SVM | CSI Amplitude and phase | 86.90% | It realizes efficient recognition of fine-grained sign language gestures. | There is no discussion of two-handed sign language gestures, and the recognition performance of the model is different in different scenarios. |

It can be seen from the specific information of various methods in Table 2 that although the above six methods have good recognition effects, the recognition accuracy of Wi-CAL in the six action division areas is better, and Wi-CAL is more suitable for scene switching in practical applications. In summary, Wi-CAL can meet the needs of action recognition in general situational perception space as a whole and can ensure stable recognition accuracy.

## 4. Conclusions

This research focuses on the wider applicability of the model. Due to the differences in motion CSI in different scenes, the effect of model recognition is also different. Therefore, we proposed a motion recognition method based on CORAL LOSS to solve the applicability of models in different scenes. At the same time, the feature selection technology ReliefF and the data enhancement technology DBA were used to further improve the model recognition rate. Several groups of experiments proved that the model we proposed could have not only good recognition ability for training and testing action data but also be applicable to the same motion data in other scenes, and it also has good classification ability for other similar public data sets. Therefore, we really realized the scene-independent action recognition model. Compared with some benchmark methods (including common machine learning and deep learning classification algorithms, etc.), Wi-CAL still has good advantages in recognition accuracy. Next, we need to: (1) improve the recognition accuracy of this model for other types of data sets, (2) further improve the environment and personnel migration ability, and further improve the generalization ability of the model.

**Author Contributions:** Z.H. contributed to conceiving and designing the experiments and revised and edited the manuscript, Z.H. and X.D. conceptualized and theorized research; J.N. and D.F. performed the experiments; Z.H. and J.N. analyzed the data and wrote the original draft; Z.H. reviewed and edited the paper; X.D. contributed significantly to revise the paper; D.F. helped in writing the related works. All authors have read and agreed to the published version of the manuscript.

**Funding:** This work was funded by Gansu Province Science and Technology Support Key R & D Program Project (20YF8GA048), 2019 Chinese Academy of Sciences "Light of the West" Talent Program, 2019 Lanzhou science and technology plan project (2019-4-44), 2020 Lanzhou talent innovation and Entrepreneurship Project (2020-RC-116) and Gansu Provincial Department of Education: industrial support plan project (2022CYCZ-12).

**Data Availability Statement:** Data are contained within the article.

**Conflicts of Interest:** The authors declare no conflict of interest.

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
