# Peer review of "Wi-CAL: A Cross-Scene Human Motion Recognition Method Based on Domain Adaptation in a Wi-Fi Environment"

_electronics, doi:10.3390/electronics11162607_

Round 1

Reviewer 1 Report

In this paper, the authors first perform data augmentation based on Dynamic Time Warping Barycentric Averaging (DBA). Then, the feature weighting algorithm and convolution layer are combined to pay more attention to action features, which selects the most representative CSI data features. Finally, a classification model for multiple scenarios is obtained by mixing softmax classifier and correlation alignment (CORAL) loss. Here, I have several concerns that I believe should be addressed.

Comments:

1. Please use the full name in the section title, such as section 2.1

2. Section 2 is not related work, but more like a technical background introduction. I may suggest the authors merge it into the method section.

3. If equations are not original, please give a reference

4. At the end of Section 3, an explanation of the recognition results needs to be given. There is totally 3 results output in Figure 6, but it should add an introduction about how to obtain the final recognition results.

5. Section 4 is a confusing expression about how many number of persons are in the experiment. Line 353 mentions that 10 people participated in the experiment, but line 378 says that it is 6 people. Please give a more accurate introduction.

6. Section 4 is a confusing expression about how many people were in the experiment. Line 353 mentions that 10 people participated in the experiment, but line 378 says that it is 6 people. Please give a more accurate introduction.

7. The output comparison of accuracy, recall and precision in Figure 9 is not accepted. For the display of the algorithm performance results, please add a simpler and easier-to-understand ROC curve and PR curve, bar charts are not a good option. The chart shows that 4 sub-graphs can be used for A, B, C and D. F1 values can be stated in the table and AUC values can be added to the ROC curve.

8. The problem in Figure 10 is similar to Figure 9, please use the ROC curve and PR curve.

9. The X-axis in Figure 12 needs to be reversed, please sort from small to large (that is, start from 5ms, not 20ms)

10. The explanation of 20ms in Line 462 is far-fetched and cannot give a reasonable explanation, why the recognition effect of Wi-CAL is usually reasonable when the sampling interval does not exceed 20ms?

11. In Figure 14, please use the accuracy curve to show the performance of the algorithm, a simple bar chart is not a good way to show the algorithm performance.

12. The author gives the comparison results of different classifier algorithms in Figure 14. However, it is recommended that the author add the other related work results in the Human Activity Recognition field for the comparison table. Instead of just comparing their own experimental results. For example, the author can use the SOTA comparison method. 

13. All the results are derived from the experimental output, and the paper needs to be expanded to describe the results compared with similar work by others in the Human Activity Recognition field. Finally, it should discuss the advantages and disadvantages of the proposed algorithm. 

Reviewer 2 Report

The authors built a model called Wi-CAL for a cross-scene human recognition. Some notes to the authors are listed below:

1. The conclusions need to be rewritten, it has unnecessary, overused phrases. 

2. in section 4.2.7 add references to the compared methods. 

3. Figures 8, 10, and 13 are blurred, please improve them.

4. Figure 10, need to explain more the difference between a and b. 

Round 2
